# Bromelain Confers Protection Against the Non-Alcoholic Fatty Liver Disease in Male C57BL/6 Mice

**DOI:** 10.3390/nu12051458

**Published:** 2020-05-18

**Authors:** Po-An Hu, Chia-Hui Chen, Bei-Chia Guo, Yu Ru Kou, Tzong-Shyuan Lee

**Affiliations:** 1Graduate Institute and Department of Physiology, College of Medicine, National Taiwan University, Taipei 10051, Taiwan; d07441003@ntu.edu.tw (P.-A.H.); chiahui1993@gmail.com (C.-H.C.); d07441002@ntu.edu.tw (B.-C.G.); 2Department of Physiology, School of Medicine, National Yang-Ming University, Taipei 10051, Taiwan; yrkou@ym.edu.tw

**Keywords:** bromelain, non-alcoholic fatty liver disease, lipoprotein metabolism, bile acid metabolism, β-oxidation

## Abstract

We aimed to investigate the effect of bromelain, the extract from stems of pineapples on the high-fat diet (HFD)-induced deregulation of hepatic lipid metabolism and non-alcoholic fatty liver disease (NAFLD), and its underlying mechanism in mice. Mice were daily administrated with HFD with or without bromelain (20 mg/kg) for 12 weeks, and we found that bromelain decreased the HFD-induced increase in body weight by ~30%, organ weight by ~20% in liver weight and ~40% in white adipose tissue weight. Additionally, bromelain attenuated HFD-induced hyperlipidemia by decreasing the serum level of total cholesterol by ~15% and triglycerides level by ~25% in mice. Moreover, hepatic lipid accumulation, particularly that of total cholesterol, free cholesterol, triglycerides, fatty acids, and glycerol, was decreased by 15–30% with bromelain treatment. Mechanistically, these beneficial effects of bromelain on HFD-induced hyperlipidemia and hepatic lipid accumulation may be attributed to the decreased fatty acid uptake and cholesteryl ester synthesis and the increased lipoprotein internalization, bile acid metabolism, cholesterol clearance, the assembly and secretion of very low-density lipoprotein, and the β-oxidation of fatty acids by regulating the protein expression involved in the above mentioned hepatic metabolic pathways. Collectively, these findings suggest that bromelain has therapeutic value for treating NAFLD and metabolic diseases.

## 1. Introduction

Hepatic lipid metabolism plays a crucial role in maintaining whole-body lipid homeostasis, including lipid production and lipid clearance with fluxes in dietary, and lipid pools of circulation and peripheral tissues [1,2,3]. Under physiological conditions, the level of lipids in liver is tightly regulated by the integration of several aspects of metabolic pathways, including *de novo* lipogenesis, the β-oxidation of fatty acids, very-low density lipoprotein (VLDL) assembly and secretion, lipoprotein internalization from circulation and bile acid synthesis [4,5,6]. The deregulation of hepatic lipid metabolism results in the initiation and progression of non-alcoholic fatty liver disease (NAFLD) [7]. NAFLD is the most common liver disease in the world and is characterized by excess fat accumulates in the liver due to the causes other than alcohol abuse [8,9,10]. It includes a broad spectrum of disorders, containing, non-alcoholic steatohepatitis to severe manifestations, such as fibrosis and cirrhosis [11]. The prevalence of NAFLD is highly associated with western-style diet, obesity and atherosclerosis [10,12,13]. Therefore, NAFLD is considered a key risk factor for the mortality both hepatic and cardiovascular diseases [12,13]. Despite the fact that several therapeutic strategies have been suggested [14,15], however, the medical treatment of NAFLD is not optimal yet.

Bromelain, the main active component extracted from the pineapple stems, has several beneficial effects, including anti-inflammation, anti-coagulation and anti-tumor growth in human and experimental animals [16,17,18]. The pharmacokinetics of bromelain in human has been performed and found that after oral administration (3 g/day) for 3 days, the non-degraded bromelain can get into circulation by intestinal absorption and retain its biological enzyme activity [19]. The plasma concentration of bromelain could reach as much as ~4000 pg/mL by 48 h and then declined to the undetectable level at 96 h after bromelain treatment [19]. Furthermore, it has been used to treat osteoarthritis and prevent swelling and inflammation after surgery [20,21]. In addition to its anti-inflammatory property, bromelain was found to have anti-obesity action [22]. With bromelain treatment, the adipogenesis of 3T3L1 cells is inhibited; in contrast, the tumor necrosis factor-α-induced lipolysis and the apoptosis of mature adipocytes were induced, and these key events may work in concert to decrease adiposity and obesity [22]. The homeostasis of whole-body lipid metabolism is tightly controlled by the liver and the crosstalk with adipose tissues and other organs under physiological condition [1,2,3,23]. However, little is known about the effect of bromelain on the regulation of hepatic lipid metabolism and the pathogenesis of NAFLD.

In this study, we aimed to clarify the interaction between bromelain effects and the high-fat diet (HFD)-induced deregulation of hepatic lipid metabolism and NAFLD. We first investigated the effect of bromelain on body weight, organ weight and the lipid profile of plasma in HFD-fed wild-type (WT) mice and then examined whether bromelain affected the HFD-induced deregulation of lipid metabolism and its underlying molecular mechanisms in the liver. Our findings demonstrate that bromelain decreased fatty acid uptake and cholesteryl ester synthesis and promoted lipoprotein internalization, bile acid metabolism, cholesterol clearance, VLDL assembly and secretion, and the β-oxidation of fatty acids in the liver, and led to the prevention of HFD-induced obesity, hyperlipidemia and the progression of fatty liver.

## 2. Materials and Methods

### 2.1. Reagents and Antibodies

Bromelain was from Cayman Chemical (Ann Arbor, MI, USA). Mouse antibodies for ATP-binding cassette A1 (ABCA1, ab18180), carnitine palmitoyltransferase 1 (CPT1α, ab128568) and liver X receptor α (LXRα, ab41902); Rabbit antibodies acyl-CoA:cholesterol acyltransferases 1 (ACAT1, ab168342), acetyl-CoA carboxylaseα (ACCα, ab109368), acyl-coenzyme A oxidase 1 (ACOX1, ab59964), apoB (ab31992), lipoprotein receptor-related protein 1 (LRP1, ab92544), peroxisome proliferator-activated receptor gamma coactivator 1-α coactivator-1α (PGC-1α, ab54481) and scavenger receptor class B type I (SR-BI, ab52629) were from Abcam (Cambridge, UK). Mouse antibodies for sterol regulatory element-binding protein (SREBP)-1 (557036) and SREBP-2 (557037) were from BD Bioscience (San Jose, CA, USA). The mouse antibody for β-actin (A5441) was from Sigma-Aldrich (St Louis, MO, USA). The rabbit antibody for apoE (1930-5) was from Epitomics (Burlingame, CA, USA). The rabbit antibody for low-density lipoprotein receptor (LDLR, NB110-57162) was from Novus Biologicals (Littleton, CO, USA). The rabbit antibody for apoAI (14427-1-AP) was from Proteintech (Rosemont, IL, USA). The mouse antibody for peroxisome proliferator-activated receptor γ (PPARγ, sc-7273); Rabbit antibodies for ABCG5 (sc-25796), ABCG8 (sc-30111), long-chain fatty acid coenzyme A ligase 1 (ACSL1, sc-98925), cluster of differentiation 36 (CD36, sc-9154) cytochrome P450 family 7 subfamily A member 1 (CYP7A1, sc-25536), diacylglycerol O-acyltransferase 1 (DGAT1, sc-32861), fatty acid synthase (FAS, sc-20140), hydroxymethylglutaryl-CoA reductase (HMGCR, sc-33827), liver-type fatty acid binding protein (L-FABP, sc-50380), LXRβ (sc-1001), PPARα (sc-9000) and retinoid X receptor (RXR, sc-774); goat antibodies for ACAT2 (sc-30279), DGAT2 (sc-32400) and microsomal triglyceride transfer protein (MTP, sc-33116) were from Santa Cruz Biotechnology (Santa Cruz, CA, USA). ELISA kits for cytokines and adipokines were from R&D systems (Minneapolis, MN, USA). Total cholesterol and cholesteryl ester, triglyceride, fatty acid and glycerol fluorometric assay kits were obtained from BioVision (Milpitas, CA, USA).

### 2.2. Animals

This study conformed to the Guide for the Care and Use of Laboratory Animals (Institute of Laboratory Animal Resources, eighth edition, 2011), and all animal experiments were approved by the Animal Care and Utilization Committee of National Yang-Ming University (No. 1070314). Male wild-type (WT) C57BL/6 mice at 8 weeks old were obtained from National Laboratory Animal Center (Taipei, Taiwan) and were randomly divided into two groups and administrated with either vehicle (PBS) or bromelain (20 mg/kg) by oral gavage with high-fat diet (HFD, 60% fat) for 12 weeks. Mice were housed in barrier facilities, maintained in a 12-h/12-h light-dark cycle. Temperature (22 °C) and humidity (40–60%) of the vivarium were tightly controlled. At the end of the experiment, mice were euthanized with CO_2_. The liver, white adipose tissue (WAT) and brown adipose tissue (BAT) were isolated and weighed. These samples were then subjected to histological analysis or stored at −80 °C. The frozen liver, WAT, and BAT were homogenized, and then the lysates were subjected to Western Blot analysis.

### 2.3. Histology Examination

The liver blocks were cut into 8 μm sections and subjected to histological examination. Cryosections were subjected to oil red O staining and deparaffinized sections were stained with hematoxylin and eosin (H&E), and then viewed under a Motic TYPE 102M microscope (Motic Images, Xiamen, China).

### 2.4. Western Blot Analysis

The livers were lysed in immunoprecipitation lysis buffer (50 mM Tris pH7.5, 5 mM EDTA, 300 mM NaCl, 1% Triton X-100, 1 mM phenylmethylsulfonyl fluoride, 10 μg/mL leupeptin, 10 μg/mL aprotinin and phosphatase inhibitor cocktail ΙΙ and ΙΙΙ). Aliquots of tissue lysates (50 μg protein) were mixed with 5 μL loading dye (250 mM Tris HCl pH6.8, 500 mM dithiothreitol, 10% SDS, 50% glycerol and bromophenol blue), and then boiled at 100 °C for 5 min. Proteins were separated on 8%, 10% or 15% SDS-polyacrylamide gel by electrophoresis. The proteins were separated and transferred onto a polyvinylidene fluoride membrane (Pall, NY, USA), blocked with 5% skim milk for 1 h at room temperature, and then incubated with primary antibodies overnight and then with corresponding secondary antibody for 2 h. The protein bands were detected by using an enhanced chemiluminescence kit (PerkinElmer, Boston, MA, USA) and quantified using ImageQuant 5.2 (Healthcare Bio-Sciences, Pittsburgh, PA, USA).

### 2.5. Serum Lipid Analysis

After euthanizing with CO_2_, the blood was collected by cardiac puncture. After clotting and centrifugation, serum was isolated and the levels of triglycerides, total cholesterol, HDL-cholesterol (HDL-c), and non-HDL-c were evaluated by using the corresponding test strips of Spotchem EZ SP 4430 (ARKRAY, Inc., Kyoto, Japan).

### 2.6. Hepatic Lipid Measurement

The lipids in the liver were extracted by use of isopropanol/chloroform/NP-40 (11:7:0.1, *v/v*). After drying, the extracted lipids were dissolved by using the commercial solutions and the levels of total cholesterol, free cholesterol, cholesteryl ester, triglycerides, fatty acids, and glycerol were measured by using fluorescence assay kits (BioVision, Milpitas, CA, USA). The fluorescence intensity was detected at 535 nm excitation and 590 nm emission under the multimode microplate reader Infinite 200 (TECAN, Männedorf, Switzerland).

### 2.7. Measurement of Cytokines and Adipokines

The levels of pro-inflammatory cytokines and adipokines in WAT, including tumor necrosis factor-α (TNF-α), monocyte chemoattractant protein-1 (MCP-1) and interleukin-6 (IL-6), and adiponectin, leptin, and resistin were measured by using ELISA kits according to the manufacturer’s instructions.

### 2.8. Statistical Analysis

The results were presented as mean ± SEM from 10 mice. Data from mice were evaluated by parametric tests. The unpaired *t*-test was used to compare two independent groups. SPSS software v20.0 (Chicago, IL, USA) was used for analysis. Differences were considered statistically significant at *p* < 0.05.

## 3. Results

### 3.1. Effects of Bromelain on Body Weight, Organ Weight and Blood Lipid Profile of Mice Upon HFD Administration

Treatment with bromelain did not affect the appetite of mice. There was no difference in food intake daily between the vehicle group and bromelain group (vehicle group: bromelain group = 4.16 ± 0.25 g: 4.23 ± 0.38 g/mouse). We demonstrated that daily treatment of mice with HFD bromelain for 12 weeks altered HFD-induced change in their appearance (Figure 1A), body weight, organ weight, and their ratio of organs weight to body weight, as compared to vehicle-treated C57BL/6 mice (Figure 1B–J). Chronic treatment with bromelain decreased the cell size of adipocytes in WAT and BAT, and leukocyte infiltration in WAT (Appendix A). In addition, bromelain increased the level of leptin and decreased the levels of MCP-1, IL-6 and resistin but not TNF-α and adiponectin in WAT (Appendix A). Moreover, treatment with bromelain decreased the circulating levels of total cholesterol, triglycerides, and non-HDL-c; in contrast, it increased the levels of HDL-c and free fatty acids in blood of C57BL/6 mice (Figure 1K–O).

### 3.2. Effects of Bromelain on HFD-Induced Lipid Accumulation in the Liver

The liver is the most important organ for the homeostasis of whole-body lipid metabolism [1,24]. We then examined the effect of bromelain on the pathogenesis of HFD-induced NAFLD. The results of gross lesions, H&E and oil red O staining showed that the daily treatment of mice with bromelain for 12 weeks significantly attenuated lipid accumulation (Figure 2A–C). In addition, treatment with bromelain decreased hepatic levels of total cholesterol, free cholesterol, triglycerides, free fatty acids, and glycerol but not cholesteryl ester (Figure 2D–I). These results suggest that decreased levels of lipid accumulation in the liver due to bromelain treatment might be attributed to the reduction in total cholesterol, free cholesterol, triglyceride levels, free fatty acids, and glycerol.

### 3.3. Effects of Bromelain on the Metabolic Pathways of Lipids in the Liver

The lipid regulation in the liver is tightly controlled by several metabolic pathways, including *de novo* lipogenesis, lipoprotein metabolism, bile acid metabolism, triglyceride metabolism, the assembly and secretion of VLDL, and the β-oxidation of fatty acids [25,26,27,28]. We next delineated the mechanisms underlying the beneficial effect of bromelain on HFD-induced lipid accumulation in the liver by assessing the protein expression of hepatic lipid metabolic pathways. As compared to vehicle treatment, daily bromelain treatment increased the protein expression of SREBP-1 and SREBP-2, as well as their downstream proteins ACCα, FAS, LDLR and HMGCR (Figure 3A–I), all of which are the key regulators for fatty acid synthesis and cholesterol synthesis [8,11].

Additionally, treatment with bromelain increased the protein expression of LRP-1, LDLR and SR-BI (Figure 4A–D), the crucial regulators for the internalization of VLDL, LDL and HDL from circulation, respectively [29,30]. Furthermore, bromelain decreased the protein expression of ACAT-1 and ACAT-2 (Figure 4A,E,F), the two key enzymes for the synthesis of cholesteryl ester in the liver [31,32]. Notably, bromelain also upregulated the protein expression of LXRα, ABCA1, apoAI, CYP7A1, ABCG5 and ABCG8 (Figure 4G–L), all of which are involved in the hepatic cholesterol clearance pathway, including HDL biogenesis and bile acid metabolism [33,34,35,36]. These results suggest that bromelain may have a lipid-lowering effect and promote the cholesterol clearance from the liver (Figure 4M).

Moreover, treatment with bromelain decreased the protein expression of L-FABP and CD36 (Figure 5A–C), two crucial hepatic regulators in the internalization of fatty acids from circulation [37]. Additionally, bromelain increased the protein expression of DGAT2 and DGAT1 (Figure 5A,D,E), the key regulators of triglycerides synthesis in the liver [38]. Treatment with bromelain increased the protein expression of apoB, apoE and MTP (Figure 5F–G), the key players in the process of VLDL assembly and secretion [39,40]. These results suggest that bromelain may inhibit fatty acid uptake and promote the assembly and secretion of VLDL and thereby lead to the decreased lipid accumulation by HFD in the liver (Figure 5I).

### 3.4. Bromelain Endorses Catabolic Metabolism of Fatty Acids in the Liver

The β-oxidation of fatty acids is also the key metabolic process in regulating the intracellular lipid content of hepatocytes [5]. Thus, we next evaluated the effects of bromelain on the expression of proteins related to β-oxidation of fatty acids in the liver. We demonstrated that bromelain strikingly increased the expression of proteins involved in β-oxidation of fatty acids, including ACSL1, ACOX-1, CPT1α, RXR, PPARα, PPARγ, and PGC1α (Figure 6A–I). These findings suggest that bromelain may increase catabolic metabolism of fatty acids in the liver.

## 4. Discussion

Bromelain is known to have beneficial effects on inflammatory diseases [21]. However, the significance of bromelain on metabolic diseases is unclear. In this study, we provided new insights into the effect of bromelain and its underlying molecular mechanisms on the HFD-induced deregulation of hepatic lipid metabolism, hyperlipidemia and NAFLD in male C57BL/6 mice. We used the HFD-fed mouse model to investigate the effect of bromelain on hepatic lipid metabolism and found that chronic treatment with bromelain for 12 weeks decreased the HFD-induced increase in the body weight and the organ weight, including WAT, BAT and the liver, and attenuated hyperlipidemia and lipid accumulation in the liver of male C57BL/6 mice. Additionally, bromelain largely decreased the level of glycerol and the lipid accumulation in the liver by HFD, particularly that of cholesterol and triglycerides and free fatty acids. However, the information about the involving mechanisms behind the beneficial effect of bromelain on hepatic lipid metabolism is limited. We then explored the effect of bromelain on these lipid metabolic pathways in the HFD-fed mice.

Intriguingly, treatment with bromelain increased the protein expression related to the *de novo* lipogenesis of cholesterol and fatty acid biosynthesis such as SREBP1 and 2, FAS, ACC, HMGCR in the liver. The previous reports have demonstrated that increased *de novo* lipogenesis contributes to the development of NAFLD and that the genetic overexpression of regulators involved in *de novo* lipogenesis exacerbates the NAFLD [4,41,42]. These findings seem to be contradictory to our observation that bromelain prevented the HFD-induced hyperlipidemia and lipid accumulation in the liver. Importantly, in addition to the *de novo* lipogenesis, the homeostasis of lipid metabolism in the liver is precisely controlled by several aspects of metabolic pathways, including lipoprotein metabolism, cholesterol metabolism, bile acid metabolism and fatty acid metabolism under physiological condition or pathological situations [25,27,28]. Therefore, the molecular mechanism underlying the beneficial effect of bromelain on the HFD-induced deregulation of hepatic lipid metabolism remains to be investigated.

Of particular interest is our finding that treatment with bromelain increased the protein levels of lipoprotein receptors LRP1, LDLR and SR-BI, which might contribute to the decreased serum levels of cholesterol and triglycerides by bromelain. These results are in agreement with the previous reports that the genetic overexpression of these receptors inhibits the development of hyperlipidemia and NAFLD in mice [43,44,45]. The lipid-lowering effect of bromelain may be attributed to the up-regulation of lipoprotein receptors for uptaking lipoproteins from the circulation. However, treatment with bromelain decreased the protein expression of ACAT1 and ACAT2, two key regulators in the synthesis of cholesteryl ester in the liver [31,32]. In contrast, bromelain upregulated the protein expression of LXRα, ABCA1, apoAI, CYP7A1, ABCG5 and ABCG8, the regulators involved in the cholesterol clearance-, HDL assembly- and bile acid metabolism-related pathways [33,34,35,36,46]; all of these key events are crucial for decreasing the HFD-induced cholesterol accumulation in the liver. Collectively, these results suggest that bromelain may inhibit the cholesteryl ester biosynthesis and increase lipoprotein internalization from circulation and promote bile acid metabolism, HDL biogenesis and cholesterol clearance from the liver, which, in turn, conferred protection from the HFD-induced hyperlipidemia and the deregulation of hepatic cholesterol metabolism.

Regarding with the fatty acid metabolism in the liver, we found that treatment with bromelain decreased the protein expression of L-FABP and CD36, two important regulators for fatty acid uptake from circulation into hepatocytes; however, it increased the protein expression of DGAT1 and DGAT2, the two key enzymes for the biosynthesis of triglycerides [38]. Interestingly, bromelain treatment upregulated the protein expression of apoB, apoE and MTP, all of which are involved in the process of VLDL assembly and secretion [39,40]. These results suggest that bromelain may promote the assembly and secretion of VLDL and lead to a decrease in hepatic level of triglycerides, which is consistent with the findings of van de Sluis et al. and de Beer et al. who found that an increase in the function of VLDL assembly and secretion ameliorated the development of NAFLD in mice [29,30]. These findings suggest that bromelain may limit the influx of fatty acids, the resource for triglycerides synthesis, and accelerates the lipid efflux by promoting VLDL assembly and secretion into circulation, leading to the decrease in HFD-induced lipid accumulation in the liver.

In particular, we demonstrated that bromelain upregulated the expression of proteins related to ACSL1, ACOX1, CPT1α, RXR, PPARα, PPARγ, and PGC-1α, all involved in the β-oxidation pathway for bioenergetics in mitochondria [47,48,49,50,51]. These findings suggest that bromelain promoted the function of β-oxidation pathways in the catabolic metabolism of fatty acids, leading to the decrease in HFD-induced lipid accumulation in the liver. PPARγ is known to be upregulated in hepatic steatosis [52]. Under the NAFLD environment, the up-regulation of PPARγ is suggested to improve insulin sensitivity and glucose homeostasis, and further promote the expression of CD36 to increase the uptake of fatty acids from circulation [52,53]. Additionally, the activation of PPARγ ameliorates hepatic steatosis by increasing serum adiponectin and upregulating the gene expression related to the β-oxidation pathway [54]. In this study, we found that bromelain upregulated the protein expression of PPARγ in the liver of HFD-fed mice. We thought that the increased level of PPARγ may contribute to the upregulation of β-oxidation pathway-related proteins by bromelain. This observation agrees with previous reports that an increase in fatty acid oxidation enhances energy metabolism and retards the development of NAFLD [42,55]. In view of the function of these proteins in β-oxidation pathways, bromelain might maintain the lipid homeostasis in the liver by promoting the catabolic metabolism of lipid droplets and energy metabolism deregulated by HFD.

Specifically, autophagy is a conserved self-eating process that is important for lipid homeostasis under various stress conditions [56,57]. It allows cells to self-degrade lipid droplets in lysosomes for balancing energy sources upon nutrient shortage or various pathological insults [58,59]. Notably, our findings suggest that the decreased level of hepatic triglycerides might be attributed to the increased hydrolysis of triglycerides, as evidenced by the increased serum level of free fatty acids. Autophagy is known to stimulate the hydrolysis of triglycerides to fatty acids and attenuate the progression of NAFLD [57]. However, whether bromelain regulates the turnover of triglycerides to fatty acids is largely unknown. Nevertheless, the molecular mechanism behind the catabolic metabolism of lipid droplets to fatty acids contributing to bromelain-mediated protection in the hepatic lipid homeostasis requires further investigation.

The crosstalk between the liver and adipose tissues is crucial for maintaining the homeostasis of whole-body lipid metabolism [3,60]. It is well established that the deregulation of lipid metabolism in adipose tissues is highly associated with the metabolic diseases [1,61]. Interestingly, we also found that treatment with bromelain attenuated the HFD-induced obesity and deregulated adiposity in WAT and BAT, as evidenced by the decreasing size of adipocytes and the BAT whiting (Appendix A). In addition, bromelain inhibited the inflammatory response within WAT, as evidenced by the decreased levels of leukocyte infiltration and pro-inflammatory cytokines MCP-1, IL-6 and resistin (Appendix A). Interestingly, our findings from chow diet-fed mice showed that bromelain decreased the body weight, the weight of white adipose tissue and hepatic levels of cholesteryl ester without changing the serum level of total cholesterol and triglycerides; in contrast, bromelain increased the weight of brown adipose tissue, the serum levels of HDL-c and fatty acids and hepatic levels of non-esterified cholesterol and fatty acids in chow diet-fed mice (Appendix A). These results strongly suggest that bromelain has a regulatory effect on the lipid metabolism of mice under normal conditions. Nevertheless, how bromelain regulates the adiposity of WAT and BAT is still elusive. To this end, further investigations describing the implications of bromelain in the lipid metabolism of adipose tissues and related molecular mechanism are warranted.

## 5. Conclusions

In spite of the unique pathways discovered in this study, the detailed mechanisms of bromelain to affect lipid metabolism merit further study. Overall, congruous data have been reported, suggesting the alteration in protein expression of these key molecules regulating lipid metabolism, which again demonstrates the beneficial effects of bromelain on lipid metabolism. In conclusion, our study provides new evidence that bromelain has beneficial effects against the HFD-induced deregulation of hepatic lipid metabolism by decreasing fatty acid internalization and promoting VLDL secretion, lipoprotein metabolism, HDL assembly, cholesterol clearance and fatty acid β-oxidation in male C57BL/6 mice. These key events might work in concert to decrease the HFD-induced lipid accumulation in the liver (Figure 7).

Moreover, we discovered a link between bromelain and hepatic lipid metabolism, which broadens its biomedical impact in the treatment of metabolic diseases.

## Figures and Tables

**Figure 1 nutrients-12-01458-f001:**
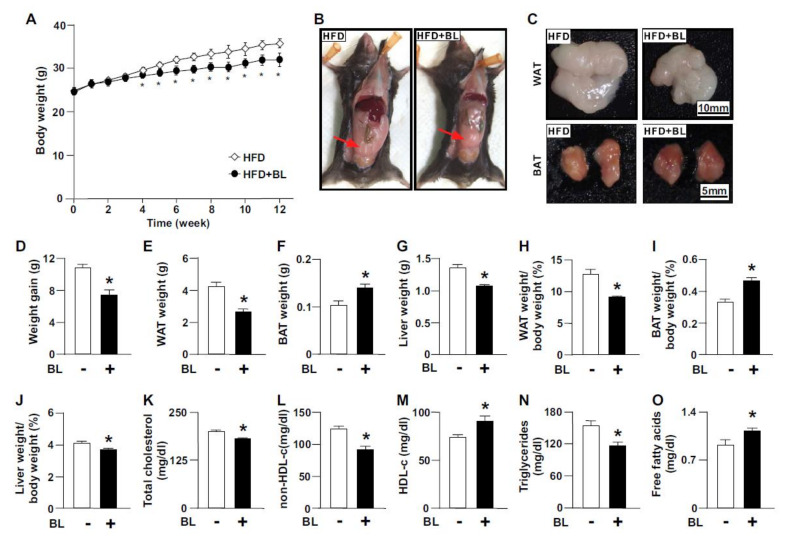
Effects of bromelain on body weight, tissue weight, and blood lipids of high-fat diet (HFD)-fed mice. With the HFD, eight-week-old C57BL/6 mice were treated daily with bromelain (20 mg/kg) or vehicle (PBS) for 12 weeks. (**A**) The time-dependent change in body weight. (**B**,**C**) The images of body appearance and white adipose tissue (WAT) and brown adipose tissue (BAT). In (**B**), WAT is indicated by arrows. (**D**) The weight gain after HFD treatment with or without bromelain. (**E**–**G**) The organ weights of WAT, BAT and the liver. (**H**–**J**) The ratios of organ weight to body weight in WAT, BAT and the liver. (**K**–**O**) Serum levels of total cholesterol, non-high-density lipoprotein cholesterol (non-HDL-c), HDL cholesterol (HDL-c), triglycerides, and free fatty acids. The results are presented by the mean ± SEM from 10 mice. * *p* < 0.05 vs. vehicle group. BL: bromelain.

**Figure 2 nutrients-12-01458-f002:**
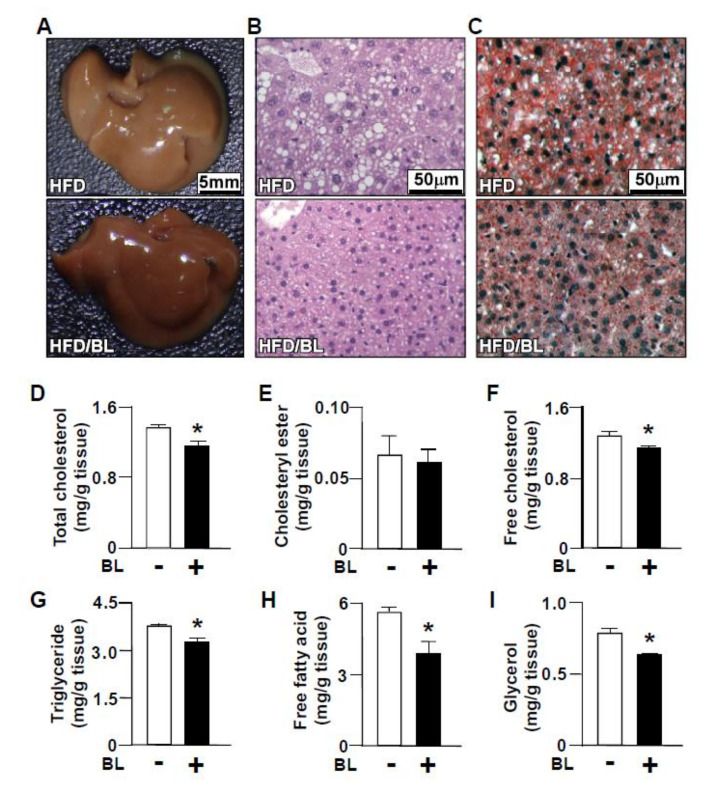
Bromelain decreases HFD-induced lipid accumulation in the liver of HFD-fed mice. Eight-week-old C57BL/6 mice were fed with HFD and were treated daily with bromelain (20 mg/kg) or vehicle (PBS) for 12 weeks. (**A**–**C**) The liver appearance and representative histological images by hematoxylin and eosin (**H**,**E**) staining and oil red O staining of the liver tissues. (**D**–**I**) The hepatic levels of total cholesterol, free cholesterol, cholesteryl ester, triglycerides, free fatty acids, and glycerol. The results are presented by the mean ± SEM from 10 mice. * *p* < 0.05 vs. vehicle group. BL: bromelain.

**Figure 3 nutrients-12-01458-f003:**
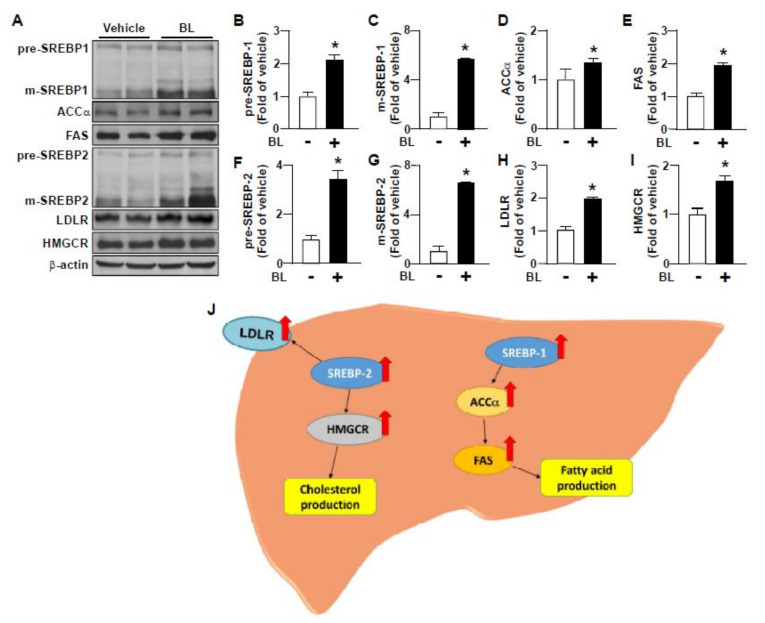
Bromelain increases *de novo* lipogenesis-related protein expression in the liver of HFD-fed mice. Eight-week-old C57BL/6 mice were fed with HFD and bromelain (20 mg/kg) or vehicle (PBS) for 12 weeks. (**A**–**I**) Western Blot analysis of protein levels of precursor of sterol regulatory element-binding proteins 1 (p-SREBP1), mature form of SREBP1 (m-SREBP1), acetyl-CoA carboxylaseα (ACCα), fatty acid synthase (FAS), pre-SREBP2, m-SREBP2, low-density lipoprotein receptor (LDLR), hydroxymethylglutaryl-CoA reductase (HMGCR), and β-actin in the liver. (**J**) Summary for the effect of bromelain on hepatic *de novo* lipogenesis. The data are represented as mean ± SEM from 10 mice. * *p* < 0.05 vs. vehicle group. BL: bromelain.

**Figure 4 nutrients-12-01458-f004:**
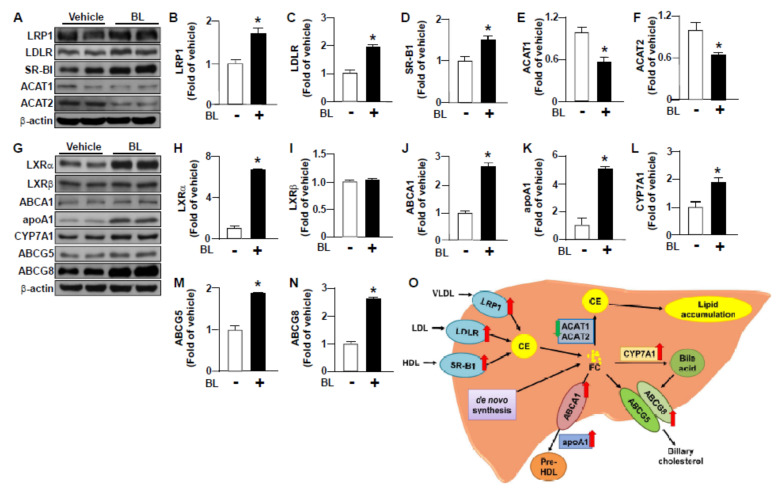
Effect of bromelain on the expression of cholesterol metabolism-related proteins in the liver of HFD-fed mice. Eight-week-old C57BL/6 mice were fed with HFD and bromelain (20 mg/kg) or vehicle (PBS) for 12 weeks. (**A**–**N**) Western Blot analysis of protein levels of lipoprotein receptor-related protein 1 (LRP1), LDLR, scavenger receptor class B type I (SR-BI), acyl-CoA:cholesterol acyltransferases 1 (ACAT1), ACAT2, liver X receptor α (LXRα), LXRβ, ATP-binding cassette A1 (ABCA1), apolipoprotein AI (apoAI), cytochrome P450 family 7 subfamily A member 1 (CYP7A1), ABCG5, ABCG8 and β-actin. (**O**) Graphic summary for the effect of bromelain on lipoprotein metabolism and bile acid metabolism in the liver. The results are presented by the mean ± SEM from 10 mice. * *p* < 0.05 vs. vehicle group. BL: bromelain.

**Figure 5 nutrients-12-01458-f005:**
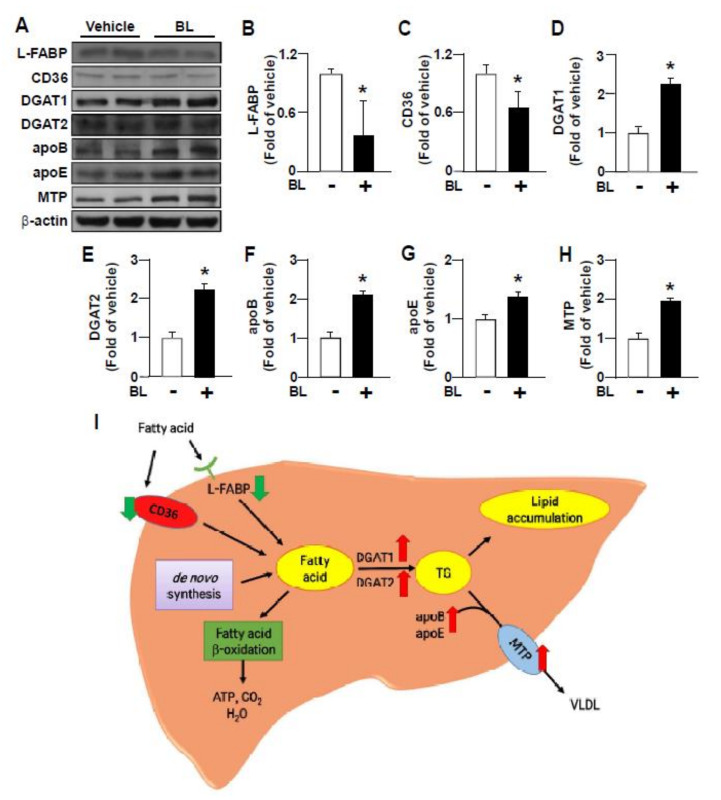
Effect of bromelain on the expression of proteins involved in the hepatic fatty acid (FA) metabolism of HFD-fed mice. Eight-week-old C57BL/6 mice were fed with HFD and bromelain (20 mg/kg) or vehicle (PBS) for 12 weeks. (**A**–**H**) Western Blot analysis of protein levels of L-FABP, CD36, DGAT1, DGAT2, apoB, microsomal triglyceride transfer protein (MTP) and β-actin. (**I**) The summary of the effect of bromelain on FA metabolism in the liver. Results are presented by the mean ± SEM from 10 mice. * *p* < 0.05 vs. vehicle group. BL: bromelain.

**Figure 6 nutrients-12-01458-f006:**
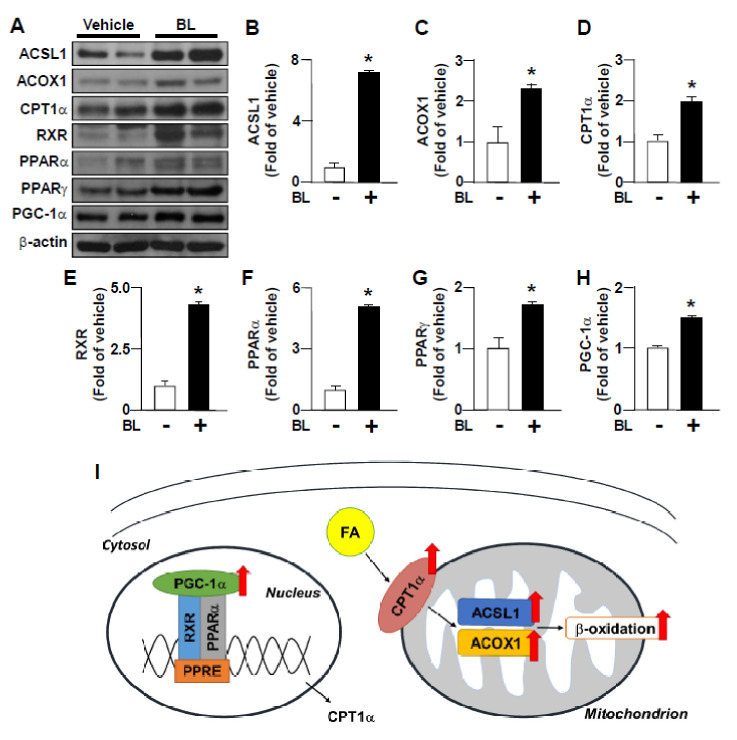
Bromelain increases the expression of FA β-oxidation-related proteins in the liver of HFD-fed mice. Eight-week-old C57BL/6 mice were fed with HFD and bromelain (20 mg/kg) or vehicle (PBS) for 12 weeks. (**A**–**H**) Western Blot analysis of protein levels of ACSL1, ACOX1, CPT1α, retinoid X receptor (RXR), PPARα, PPARγ, peroxisome proliferator-activated receptor gamma coactivator 1-α coactivator-1α (PGC-1α) and β-actin. (**I**) The summary for the effect of bromelain on FA metabolism in the liver. The results are presented by the mean ± SEM from 10 mice. * *p* < 0.05 vs. vehicle group. BL: bromelain.

**Figure 7 nutrients-12-01458-f007:**
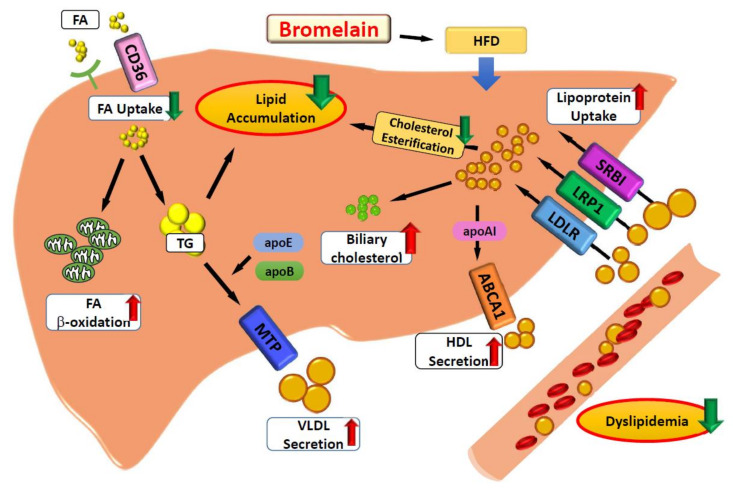
Bromelain has a beneficial effect on the deregulation of lipid metabolism by high-fat diet (HFD) in the liver by modulating multiple metabolic pathways. Bromelain prevents HFD-induced hyperlipidemia and hepatic lipid accumulation by decreasing fatty acid uptake and cholesteryl ester synthesis and promoting lipoprotein internalization, bile acid metabolism, cholesterol clearance, very-low density lipoprotein (VLDL) assembly and secretion, and the β-oxidation of fatty acids. These key events might work in concert to decrease the HFD-induced lipid accumulation in the liver.

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
