# Peer review of "Bromelain Confers Protection Against the Non-Alcoholic Fatty Liver Disease in Male C57BL/6 Mice"

_nutrients, 2020, doi:10.3390/nu12051458_

Round 1

Reviewer 1 Report

I’ve read with attention the paper of Hu et al. that is potentially of interest. The background and aim of the study have been clearly defined. The methodology applied is overall correct, the results are reliable and adequately discussed. I’ve only some minor comments:

  • The title should be modified as it follows: "Bromelain confers protection against non alcoholic fatty liver disease in C57BL/6 mice"
  • The abstract should be rewritten including more quantitative results while replacing the discussion section with a short conclusion
  • Considering Bromelain as a possible agent to be used in humans, some data on its pharmacokinetics should be added in intro or discussion
  • The authors should consider to cite the following paper: Nutrients. 2018 Aug 23;10(9). pii: E1153. doi: 10.3390/nu10091153.

Author Response

Reviewer 1

I’ve read with attention the paper of Hu et al. that is potentially of interest. The background and aim of the study have been clearly defined. The methodology applied is overall correct; the results are reliable and adequately discussed. I’ve only some minor comments:

  1. The title should be modified as it follows: "Bromelain confers protection against non-alcoholic fatty liver disease in C57BL/6 mice".

Response: We fully agree with the reviewer’s viewpoint. In response to the suggestion, we have revised the title of this manuscript in our revised manuscript (page 1, line 3).

  1. The abstract should be rewritten including more quantitative results while replacing the discussion section with a short conclusion.

Response: We thank the reviewer for the professional suggestion. In response to the suggestion, we have largely revised the abstract and added more quantitative results and replaced the discussion section with a short conclusion in our revised manuscript (page 1, line 12-27).

  1. Considering Bromelain as a possible agent to be used in humans, some data on its pharmacokinetics should be added in intro or discussion.

Response: We thank the reviewer for reminding us this important issue. In response to the suggestion, we have added the information about the pharmacokinetics of bromelain in the section of introduction in our revised manuscript. Now the sentences read as “The pharmacokinetics of bromelain in human has been performed and found that after oral administration with bromelain (3g/day) for 3 days, the non-degraded bromelain can get into circulation through intestinal absorption and retain its biological enzyme activity [19]. The plasma concentration of bromelain could reach as much as ~4000 pg/ml by 48 h and then declined to the undetectable level at 96 h after bromelain treatment [19]” (page 2, line 48-52). We sincerely hope the reviewer could approve our revision.

  1. The authors should consider to cite the following paper: Nutrients. 2018 Aug 23;10(9). pii: E1153. doi: 10.3390/nu10091153.

Response: We fully agree with the reviewer’s viewpoint. In response to the suggestion, we have revised the section of references in our revised manuscript (ref. 7-10 and 53).

Reviewer 2 Report

In this study, the authors demonstrated the role of bromelain in NAFLD. They suggested that bromelain administration could improve the lipid profile and inhibit the lipid accumulation in liver. They also investigated the lipid metabolism related genes expression in HFD induced mice liver. However, the experiment design and some results are confusing. The exact mechanism of bromelain in regulating lipid metabolism is still lacking. There are the specific issues need to be addressed.

  1. Authors should add low-fat diet group and low-fat diet bromelain treated
  2. The mice gender and food intake information should be provided, since bromelain may affect the mice appetite.
  3. Is there any evidence that bromelain could directly get into the blood to affect hepatic cell or only function through regulating intestinal digestion? Authors should discuss that.
  4. The serum AST and ALT levels should be provided.
  5. There are a lot of inconsistent results, for example, the serum and hepatic triglyceride were all decreased by bromelain administration, but SREBP1, FAS, ACC, HMGCR, LXR, apoB, PPARgamma and MTP were all significantly induced. And these results can not be neglected, only because some of fatty acid oxidation related gene were upregulated. Authors should provide more detailed mechanism study to further investigate the exact role of bromelain. And their study should mainly focus on one or two pathways, in which bromelain exert the major lipid-lowering effect.
  6. Authors do not need to add graphic summary for every single figure. A comprehensive graphic summary for the whole manuscript to demonstrate the signaling pathway of bromelain in lipid regulation is enough.

Author Response

Reviewer 2

In this study, the authors demonstrated the role of bromelain in NAFLD. They suggested that bromelain administration could improve the lipid profile and inhibit the lipid accumulation in liver. They also investigated the lipid metabolism related genes expression in HFD induced mice liver. However, the experiment design and some results are confusing. The exact mechanism of bromelain in regulating lipid metabolism is still lacking. There are the specific issues need to be addressed.

  1. Authors should add low-fat diet group and low-fat diet bromelain treated.

Response: We thank the reviewer for the professional suggestion. We indeed examined the effect of bromelain on lipid metabolism in mice with chow diet in our pilot study. Our findings that bromelain decreased the body weight, the weight of white adipose tissue and hepatic levels of cholesteryl ester without changing the serum level of total cholesterol and triglycerides; in contrast, bromelain increased the weight of brown adipose tissue, the serum levels of HDL-c and fatty acids and hepatic levels of non-esterified cholesterol and fatty acids in chow diet-fed mice. These results strongly suggested that bromelain has regulatory effect on the lipid metabolism of mice under normal condition. However, we wanted to address the protective effect of bromelain on NAFLD in this study, we therefore did not show the data from chow diet-fed mice with or without bromelain. In response to the suggestion, we have added the data of chow diet-fed mice as Supporting Information and described these results in the section discussion in our revised manuscript (page 11, line 329-335).

We sincerely hope that the reviewer can approve our arrangement for the data of chow diet-fed mice.

  1. The mice gender and food intake information should be provided, since bromelain may affect the mice appetite.

Response: We thank the reviewer for reminding us this important issue. Based on our observation, treatment with bromelain did not affect the appetite of mice. There was no difference in food intake daily among the vehicle group and bromelain group (vehicle group: bromelain group= 4.16±0.25 g: 4.23±0.38 g/mouse). In response to the suggestion, we have added the description regarding to the food intake daily of mice in the section of results of our revised manuscript (page 4, line 151-153). Regarding to the gender effect on the bromelain-conferred protection, we did not investigate the protective effect of bromelain in female mice. It has been reported that the prevalence of NAFLD is higher in men than that in women (Hashimoto and Tokushige, J Gastroenterol 2011, 46, 63–69; Ballestri et al., Adv Ther 2017, 34, 1291–1326); we therefore, only examine the effect of bromelain on the development of NAFLD in male mice.

In response to the reviewer’s suggestion, we have added the information about mouse gender in the section of Materials and Methods in our revised manuscript (page 3, line 99).

  1. Is there any evidence that bromelain could directly get into the blood to affect hepatic cell or only function through regulating intestinal digestion? Authors should discuss that.

Response: We thank the reviewer for reminding us this important issue. In response to the suggestion, we have added the information about the pharmacokinetics of bromelain in the section of introduction in our revised manuscript. Now the sentences read as “The pharmacokinetics of bromelain in human has been performed and found that after oral administration (3g/day) for 3 days, the non-degraded bromelain can get into circulation through intestinal absorption and retain its biological enzyme activity [19]. The plasma concentration of bromelain could reach as much as ~4000 pg/ml by 48 h and then declined to the undetectable level at 96 h after bromelain treatment [19]” (page 2, line 48-52). We sincerely hope the reviewer could approve our revision.

  1. The serum AST and ALT levels should be provided.

Response: We thank the reviewer for reminding us this important issue. We did not measure the serum levels of AST and ALT in this study. The serum samples from mice have been used up; we apologize for the inability to provide the data regarding to the serum levels of AST and ALT. We sincerely hope the reviewer could approve our limitation of this study.

  1. There are a lot of inconsistent results, for example, the serum and hepatic triglyceride were all decreased by bromelain administration, but SREBP1, FAS, ACC, HMGCR, LXR, apoB, PPARgamma and MTP were all significantly induced. And these results can not be neglected, only because some of fatty acid oxidation related gene were upregulated. Authors should provide more detailed mechanism study to further investigate the exact role of bromelain. And their study should mainly focus on one or two pathways, in which bromelain exert the major lipid-lowering effect.

Response: We thank the reviewer for allowing us to explain more regarding this issue. Indeed, bromelain increased the expression of proteins involved in de novo lipogenesis such as SREBP1 and 2, FAS, ACC, HMGCR. These findings seem to be contradictory to our observation that bromelain prevented the HFD-induced hyperlipidemia and lipid accumulation in the liver such as the decreased levels of the serum total cholesterol and triglycerides, hepatic free cholesterol and triglycerides. We thought that the decreased hepatic free cholesterol could result from LXRalpha-driven up-regulation of CYP7A1, ABCG5, ABCG8, ABCA1 and apoAI, all of which are crucial regulators in bile acid metabolism and cholesterol efflux to eliminate excess cholesterol from human body or promote the HDL biogenesis. Moreover, the decreased level of hepatic triglycerides may be due to the upregulation of apoB, apoE and MTP, all of which promote the assembly and secretion of VLDL, leading to a decrease in the level of triglycerides of the liver. Notably, the decreased level of hepatic triglycerides might also be attributed to the increased hydrolysis of triglycerides as evident by the increased serum level of free fatty acids. Autophagy is known to stimulate the turnover of triglycerides to fatty acids and attenuate the progression of NAFLD. However, whether autophagy involved in the bromelain-mediated protection in the hepatic lipid homeostasis and the development of NAFLD is unknown. In response to the reviewer’s suggestion, we have revised the section of discussion in our revised manuscript (page 9, line 264-266; page 10, line 269; page 10, line 285-289; page 10, line 295-296; page 10, line 310-318; page 11, line 319-320). The investigation regarding to the role of autophagy in bromelain-conferred protection in NAFLD is ongoing in our laboratory.

We sincerely hope the reviewer could approve our viewpoint and the arrangement for studying the detailed mechanism underlying the protective effect of bromelain on NAFLD.

  1. Authors do not need to add graphic summary for every single figure. A comprehensive graphic summary for the whole manuscript to demonstrate the signaling pathway of bromelain in lipid regulation is enough.

Response: We thank the reviewer for allowing us to explain more about this issue. Because the lipid metabolic pathways in the liver are quite complicated; in order to allow readers to easily understand the research results, we would like to keep the graphic summary in every single figure. We sincerely wish that the editor and the reviewer could approve our viewpoint.

Round 2

Reviewer 2 Report

Authors have improved the manuscript. They suggested that the decreased hepatic triglyceride and cholesterol were due to the elevated lipid oxidation and hepatic lipid secretion. However, they still need to explain why serum triglyceride and cholesterol levels were decreased since hepatic lipid secretion was significantly induced in their results. In addition, they should discuss the physiological significance of PPARgamma induction by bromelain in the mice liver.

Author Response

Responses to Comments of Reviewers (nutrients-788193R1)

We would like to thank the editor and the reviewer for their extensive assessment of our manuscript for their important and helpful comments/suggestions. We have taken all the remarks into account, in a manner that is described in detail below together with our answers to certain comments. We think that, following the suggestions, our revised manuscript has gained in clarity and hope that the changes made will be considered satisfactory.

Reviewer 2

Authors have improved the manuscript. They suggested that the decreased hepatic triglyceride and cholesterol were due to the elevated lipid oxidation and hepatic lipid secretion. However, they still need to explain why serum triglyceride and cholesterol levels were decreased since hepatic lipid secretion was significantly induced in their results. In addition, they should discuss the physiological significance of PPARgamma induction by bromelain in the mice liver.

Response: We thank the reviewer for reminding us this important issue. In response to the suggestion, we have discussed this point in our revised manuscript. Now the paragraph read as “our findings that treatment with bromelain increased the protein levels of lipoprotein receptors LRP1, LDLR and SR-BI, which might contribute to the decreased serum levels of cholesterol and triglycerides by bromelain. These results are agreement with the previous reports that genetic overexpression of these receptors inhibits the development of hyperlipidemia and NAFLD in mice [43-45]. The lipid-lowering effect of bromelain may be attributed to the up-regulation of lipoprotein receptors for uptaking lipoproteins from the circulation” (page 10, line 276-281).

We have also discussed the physiological significance of PPARgamma induction by bromelain in the mice liver in our revised manuscript. Now the paragraph read as “PPARgamma is known to be upregulated in hepatic steatosis [52]. Under the NAFLD environment, the up-regulation of PPARgamma is suggested to improve insulin sensitivity and glucose homeostasis, and further promote the expression of CD36 to increase the uptake of fatty acids from circulation [53,54]. Additionally, activation of PPARgamma ameliorates hepatic steatosis by increasing serum adiponectin and upregulating the gene expression related to beta-oxidation pathway [54]. In this study, we found that bromelain upregulated the protein expression of PPARgamma in the liver of HFD-fed mice. We thought that the increased level of PPAgamma may contribute to the upregulation of beta-oxidation pathway-related proteins by bromelain. This observation agrees with previous reports that increase in fatty acid oxidation enhances energy metabolism and retards the development of NAFLD [42,55]” (page 10, line 308-317). Accordingly, the reference list has also been revised.

We sincerely hope the reviewer could approve our revision.